# Noise Sources and Control, and Exposure Groups in Chemical Manufacturing Plants

**Oscar Rikhotso \*, Johannes Leon Harmse and Jacobus Christoffel Engelbrecht**

Department of Environmental Health, Tshwane University of Technology, Private Bag X680,
Pretoria 0001, South Africa

**\*** Correspondence: oscar.rikhotso@sasol.com or luvani1723@gmail.com; Tel.: +27-79-463-8771

**Abstract:** The chemical manufacturing industry employs sophisticated mechanical equipment to process feedstock such as natural gas by transforming it to usable raw material in downstream sectors. Workers employed at these facilities are exposed to inherent occupational health hazards, including occupational noise. An online and grey literature search on ScienceDirect, Oxford Journals online, PubMed, Medline, Jstor and regulatory bodies using specific keywords on noise emission sources in the manufacturing sector was conducted. This review focuses on noise sources and their control in chemical manufacturing plants along with the receptors of the emitted noise, providing hearing conservation programme stakeholders valuable information for better programme management. Literature confirms that chemical manufacturing plants operate noise emitting equipment which exposes job categories such as machine operators, process operators and maintenance personnel amongst others. Prominent noise sources in chemicals manufacturing industries include compressors, pumps, motors, fans, turbines, vents, steam leaks and control valves. Specific industries within the chemical manufacturing sector emit noise levels ranging between 85–115 dBA (A-weighted sound pressure level), which exceed the noise rating limit of 85 dBA used in South Africa and United Kingdom, as well as the 90 dBA permissible exposure level used in the United States, levels above which workplace control is required. Engineering noise control solutions for plant equipment and machinery operated in chemical manufacturing plants are available on the market for implementation.

**Keywords:** chemical manufacturing plants; noise; noise-induced hearing loss; noise source; noise exposure group; noise transmission paths

---

## 1. Introduction

The manufacturing sector inclusive of chemical manufacturing transforms materials, chemicals and components to value-added consumer and commercial goods using power-driven and material-handling machinery in installations commonly referred as plants, factories or mills [1,2]. Chemical manufacturing plants use natural gas or refinery by-products such as ethylene, hydrogen rich gas, tail gas and many more as feedstock to produce chemical products and many consumer goods in downstream chemical processes [3,4]. Plastic and paper manufacturing, petroleum refining and textile mills are but some of the industries classified under the manufacturing sector employing millions of workers worldwide [1,2].

Noise exposure in the manufacturing sector inclusive of the chemical manufacturing plants is widespread and amongst the loudest [5]. Cited reasons for the noise problem in existing installations include the inadequate knowledge of its mechanism of generation and abatement, lack of proper consideration during plant design phase and installation [6].

The noise sources operating in chemical manufacturing plants emits noise through acoustical radiation and aerodynamic turbulence [7,8]. The noise is then transmitted through pathways such

as turbulence, shock and pulsation, cavitation, impact and tooth meshing [8,9]. Noise regulations require the mandatory of implementation of hearing conservation programmes (HCPs) for exposure prevention and control in workplaces emitting noise exceeding the universal noise rating limits, 85 or 90 dBA [10–12]. However, HCPs have not eliminated excessive noise levels in some chemical industries and have tended to shift focus away from control of noise at the source towards the use of hearing protection devices (HPDs).

Exposure to occupational noise may lead to auditory and non-auditory health effects. Examples of auditory effects includes noise induced temporary threshold shift, acoustic trauma, tinnitus and noise-induced hearing loss (NIHL). Examples of the non-auditory effects include interference with speech and telephone communication, annoyance and job performance. The auditory effects of noise on humans result in a loss of hearing sensitivity whereas the non-auditory effects of noise affect the psycho-social wellbeing of humans and are mainly subjective [13,14].

In the case of compensable auditory effects, the illness affects other stakeholders such as the immediate family, co-workers, medical practitioners and insurers. In addition, there are costs associated with noise-induced illnesses relating to general economic costs, social costs, employer and employee costs. The understanding of the overall impact of noise sensitises all affected stakeholders to be aware of the true impact of noise-related illnesses and a need to plan for the implementation of required preventive actions [15,16]. Emerging issues related to noise exposure includes its effect on blood pressure, its effect on reproduction, its association with ototoxic chemicals and ototoxic medicines further highlighting the need for exposure prevention.

This review provides an appraisal of prominent noise sources and control, noise levels and exposure groups in chemical manufacturing plants.

## 2. Materials and Methods

A web based literature search on ScienceDirect, Oxford Journals online, PubMed and Jstor electronic databases was conducted up to July 2019 related to noise in the manufacturing sector and chemical manufacturing industry using the following specific keywords: noise in chemical manufacturing plants, noise in manufacturing sector, noise level database, noise sources in chemical manufacturing industry, noise control in chemical manufacturing industry, job categories in chemical manufacturing industry. The search was also extended to online databases of regulatory, standard setting and research bodies such as the National Institute for Occupational Safety and Health (NIOSH), Occupational Safety and Health Administration (OSHA), United States Environmental Protection Agency (US EPA), Health and Safety Executive (HSE), South African National Standards (SANS), World Health Organisation (WHO), International Organization for Standardization (ISO) and the International Labour Organisation (ILO). These bodies form the knowledge base in noise exposure prevention and control in industry.

The literature search was further extended to classical handbooks on occupational health and safety as well as chemical engineering books to describe process application of noise sources operated in chemical manufacturing processes. The Department of Employment and Labour (DEL) in South Africa (SA) was also searched for related content.

## 3. Results

The online literature search on databases returned limited and fragmented results. Classical handbooks on noise returned literature results closer to the search criteria for this review whilst OSHA, NIOSH (Health hazard evaluations) and the US EPA were identified as bodies having noise level records from various industries [5,17,18]. The online search on the DEL webpage only returned NIHL regulations and associated forms.

*3.1. Noise and the Manufacturing Sector*

Noise is listed as an inherent occupational hazard for processes operated within the manufacturing sector [19,20], emitted by an array of plant equipment [21]. The chemical manufacturing industry is important in the manufacturing sector as it is interconnected to downstream sectors such as transport, agriculture, power production, consumer goods and construction [1].

3.1.1. Noise Levels in the Manufacturing Sector

The diverse nature of plant equipment used by the manufacturing sector results in variance in emitted noise levels due to the age of the plant equipment, wear and tear of the equipment and operating speed of the operated machines, amongst other factors [22]. The emitted noise levels expose a great percentage of the workforce employed in the sector [23]. Table 1 shows the reported average noise levels in literature, with references, within the manufacturing sector inclusive of the chemical manufacturing industry.

From 1979 to 2013, noise measurements recorded in the OSHA database show that the majority of noise levels exceeding the permissible exposure level were from the manufacturing industry [24]. In general, data in Table 1 show that noise records from the manufacturing sector were mainly from the textile industry and emanated from different geographies. The noise records from the US date back to the 1970s, the period around which the enactment of the Occupational Safety and Health Administration Act was initiated.

There were only two noise records that emanated from SA. These records are recent and show noise levels from the chemical manufacturing industry and the iron and steel industry highlighting the need for information sharing through publications and other research platforms. In the case of SA, the low return on the literature search is understandable against the backdrop of a limited number of chemical manufacturing plants. The SA chemical manufacturing sector is anchored by a few companies concentrated close tosix refineries to enable easy access to feedstock.

The overall literature on noise for the chemical manufacturing sector retuned a low number of search results. Due to the enormous size of the sector and the important role the sector plays in the world economy, the literature search was expected to return a high volume of related literature but the expectation was proven negative. Expectedly, most of the obtained literature related to chemical manufacturing and noise emanated from the US sources. Occupational health and safety specialists employed in the chemical manufacturing sector should fill this knowledge gap through journal publications such as this review article and other platforms.

**Table 1.** Average noise levels per manufacturing industry type.

| Country | Industry | Average Noise Level in dBA [1] | References |
|---|---|---|---|
| **Food manufacturing** | | | |
| US | Food manufacturing | 90–92 | OSHA, 1979–2006 [5] US EPA, 1971 [18] |
| UK | Food processing | 88–94 | Institute of Occupational Medicine, 2002 [25] |
| **Textile plants** | | | |
| US | Textile product mills | 89–95 | OSHA, 1979–2006 [5] US EPA, 1971 [18] |
| India | Textile plants | 80–102 | Bedi, 2006 [26] |
| Sudan | **Textiles:** Weaving Preparing | 88–86 63–93 | Ahmed and Awadalkarim, 2015 [27] |
| Ethiopia | **Textiles:** Spinning mill Weaving mill | 86–115 92–101 | Ejigu, 2019 [28] |
| UK | **Textiles:** Twisting area Winding area | 88–92 82–85 | Institute of Occupational Medicine, 2002 [25] |
| South Korea | Textile plant | 81–110 | Moon and Kwon, 1976 [29] |
| US | Apparel manufacturing | 81 | OSHA, 1979–2006 [5] US EPA, 1971 [18] |
| Nigeria | Textile industry | 97–105 | Odusanya, Nwawolo, Ademuson and Akinola 2004 [30] |
| US | Leather and allied product manufacturing | 90 | OSHA, 1979–2006 [5] US EPA, 1971 [18] |
| **Wood industry** | | | |
| US | Wood product manufacturing | 92–94 | OSHA, 1979–2006 [5] US EPA, 1971 [18] |
| South Korea | Wood industry | 71–100 | Moon and Kwon, 1976 [29] |
| US | Paper manufacturing | 90–92 | OSHA, 1979–2006 [5] US EPA, 1971 [18] |
| **Printing and publishing industry** | | | |
| US | Printing and publishing | 82–93 | OSHA, 1979–2006 [5] US EPA, 1971 [18] |
| Ghana | Printing company | 79–90 | Boateng and Amedofu, 2004 [31] |

**Table 1.** *Cont.*

| Country | Industry | Average Noise Level in dBA [1] | References |
|---|---|---|---|
| **Petroleum and related industries** | | | |
| US | Petroleum and coal products manufacturing | 87–92 | OSHA, 1979–2006 [5]<br>US EPA, 1971 [18] |
| US | Refinery units:<br>Hydrocracker plant<br>Fluid cracker<br>Hydrofluoric alkylation unit<br>Catalytic hydrocracking<br>Crude distillation | 93–100<br>89–115<br>89–100<br>90–100<br>85–111 | Burgess, 1995 [21] |
| Taiwan | Oil refinery | 73–89 | Chen and Tsai, 2003 [32] |
| **Chemical manufacturing** | | | |
| SA | Chemical manufacturing company | 85-95 | Rikhotso, Harmse and Engelbrecht, 2018 [33] |
| US | Urea formaldehyde and polyurethane foam insulation:<br>Manufacturer A<br>Manufacturer B | <br><br>83–94<br>81–86 | NIOSH, 1983 [34] |
| US | Chemical manufacturing | 85–92 | OSHA, 1979–2006 [5]<br>US EPA, 1971 [18]<br>Lynch, 1989 [35]<br>Pringle and Warren, 1989 [36] |
| South Korea | Chemical industry | 91–100 | Moon and Kwon, 1976 [29] |
| US | Polyethylene battery separator manufacturing:<br>regrind<br>Extruder lines | <br>95–97<br>82–94 | NIOSH, 2004 [37] |
| Iran | Petrochemical industry | 88–93 | Neghab, Maddahi & Rafeefard, 2009 [38] |
| US | Plastics and rubber products manufacturing | 86–92 | OSHA, 1979–2006 [5]<br>US EPA, 1971 [18]<br>Mutchler, 1989 [22] |
| US | Nonmetallic mineral product manufacturing | 88–94 | OSHA, 1979–2006 [5]<br>US EPA, 1971 [18] |
| **Steel industry** | | | |
| US | Primary metal manufacturing | 91–92 | OSHA, 1979–2006 [5]<br>US EPA, 1971 [18] |
| UK | Steel industry | 90–100 | Howell, 1978 [39] |
| SA | Iron and steel companies | 78–106 | Mizan, Abrahams, Sekobe, Kgalamano et al. 2013 [40] |
| US | Fabricated metal product manufacturing | 90–92 | OSHA, 1979–2006 [5]<br>US EPA, 1971 [18] |
| Saudi Arabia | Beverage cans manufacturing<br>Steel reinforcement forming for concrete<br>Steel sheets forming and processing | 92–98<br>91–95<br>87–91 | Noweir, Bafail & Jomoah, 2014 [41] |
| Brazil | Metallurgical company | 83-102 | Guerra, Lourenco, Bustamante-Teixeira & Alves 2005 [42] |

**Table 1.** *Cont.*

| Country | Industry | Average Noise Level in dBA [1] | References |
|---|---|---|---|
| | | **Machinery manufacturing** | |
| US | Machinery manufacturing | 86–93 | OSHA, 1979–2006 [5] US EPA, 1971 [18] |
| India | Small scale hand tools manufacturing industry | 81-110 | Singh, Bhardwaj, Deepark & Bedi, 2009 [43] |
| US | Computer and electronic product manufacturing | 85–91 | OSHA, 1979–2006 [5] US EPA, 1971 [18] |
| US | Electrical equipment, appliance, and component manufacturing | 87–90 | OSHA, 1979–2006 [5] |
| Saudi Arabia | Industrial and household appliance manufacturing | 85–86 | Noweir, Bafail & Jomoah, 2014 [41] |
| US | Transportation equipment manufacturing | 88–92 | OSHA, 1979–2006 [5] US EPA, 1971 [18] |
| US | Furniture and related product manufacturing | 88–93 | OSHA, 1979–2006 [5] US EPA, 1971 [18] |
| | | **Bottling and tobacco industry** | |
| US | Beverage and tobacco product manufacturing | 86–96 | OSHA, 1979–2006 [5] US EPA, 1971 [18] |
| Nigeria | Bottling industry | 95–103 | Odusanya, Nwawolo, Ademuson & Akinola 2004 [30] |
| Nigeria | Bottling factory | 92–99 | Ologe, Olajde, Nwawolo & Oyejola, 2008 [44] |
| | | **Miscellaneous manufacturing** | |
| US | Miscellaneous manufacturing | 87–91 | OSHA, 1979–2006 [5] US EPA, 1971 [18] |

[1] A-weighted sound pressure level.

The average noise levels across the manufacturing industry ranges between 81–115 dBA inclusive of the petroleum, chemical and plastics manufacturing which are sub-categories of the chemical manufacturing sector [1]. Average noise levels noted in Table 1 which are at and/or above 85 and/or 90 dBA would require implementation of hearing conservation or control measures depending on the domicile (country) of the industry where the noise was measured [10–12]. The hearing conservation and noise control levels currently used in SA, US and the UK are shown in Table 2.

**Table 2.** Hearing conservation and control levels [45].

| Country | Permissible Exposure Level/Noise Rating Limit in dBA (8-h Average) | Exchange Rate in dB | dBA Level for HCP Institution | dBA Level for Engineering Controls |
|---|---|---|---|---|
| SA | 85 | 3 | 85 | 85 |
| UK | 80 (lower exposure action value) 85 (Upper exposure action value) 87 (exposure value with HPD use) 140 [1] (Peak noise level) | 3 | 80 | 87 |
| US | 90 | 5 | 85 | 90 |

[1] C-weighted peak noise level (dBC).

The permissible exposure levels or noise-rating limit or exposure action values listed in Table 2 are not related to the emission data that accompanies noisy equipment and machinery intended for industrial use. The objective values in Table 2 are used by regulatory bodies during enforcement and inspection activities to determine legal compliance. In SA however, the DEL inspection manual does not detail conditions under which legal compliance with the NIHL Regulations can be demonstrated. Compared to the US, the OSHA Field Operations Manual (OFM) details conditions for industry and the Certified Safety & Health Official (CSHO) under which legal compliance can be demonstrated for the noise standard.

In the UK's approach, the lower exposure value encourages employers to keep noise exposure as low as practicable, as noise levels increase between the exposure action values so does costs related to noise control. Once noise levels breach the upper exposure value, the employer will bear the costs for providing HPDs to exposed employees as well as other related costs.

In SA general industry, the 85 dBA noise rating limit is the basis for workplace control with regard to HCP initiation inclusive of training, noise measurement, area noise zoning, engineering noise control, audiometric testing and provision of HPDs.

### 3.1.2. Process Equipment Use and Noise Levels in Chemical Manufacturing Plants

Prominent and specific noise emitting process equipment in chemical manufacturing plants, whose noise is compounded by wall and ceiling reverberation include compressors, pipe fittings, pipes, pumps, control valves, flare stacks, induced draft fans and turbine generator [7,18,20,21,36,46,47]. Table 3 shows prominent process equipment and their application, noise generating mechanism and resultant noise levels in chemical manufacturing plants along with references.

**Table 3.** Process equipment, applications, noise generation mechanisms and resultant noise levels in chemical manufacturing plants.

| Process Equipment | Process Application: Noise Generation Mechanism | Specific Process Equipment Examples | Emitted Noise Level (Range) in dBA | Source or Record Type | References |
|---|---|---|---|---|---|
| Duct and pipe flow | Process product flow: High velocity flow, flow resistance, flow turbulence [9,48]. | Duct and pipe flow | 100 | Conference paper | Fagerlund, Karczub & Martin, 2005 [49] |
| Flow machines/pumps and hydraulic systems (Positive displacement and reciprocating pumps) | Pressurisation and movement of gases and fluids within pipelines: Tooth meshing, friction, inertia, rolling, cooling fan, air intake [7,9,22,47,48,50,51]. | Screw type / Vane type / Axial piston type / Gear (aluminium) type / Vane (mobile) type / Gear (machine stock) type | 71–78 / 75–82 / 76–86 / 78–88 / 84–92 / 96–104 | Handbooks | Miller, 1984 [7] / Burgess, 1995 [21] / Lynch, 1989 [35] |
| Free jets | Release of gas through nozzles. Mixing layer of the turbulence due to gas stream speed: High velocity air and steam jets [7,9,48] | Free jets (1 m from blowoff nozzle) | 105 | Handbook | Gerges, Sehrndt & Parthey, 2001 [52] |
| Valves and piping (Globe and rotary valves) | Direct control or manipulation of the process through positioning of valve plug or disc from the actuator: Cavitation, turbulence, shock and pulsation [7,53,54]. | De-areator valve / Pressurised pipes and valves / Turbine admission valve | 95–100 / 90–100 / 100 | Handbooks | Miller, 1984 [7] / Emerson, 2005 [54] |
| Fans and blowers (Axial and centrifugal fans) | Movement of high quantities of air through use of power-driven rotating impellers: Fan, speed changer, fan motor, fan shroud [7,48,51] | Forced draft fan / Induced draft fan | 100 / 90–100 | Handbook and regulatory database | Miller, 1984 [7] / US EPA, 1971 [18] |
| Compressors and turbines | Compressors (pressure generation) and Turbines (power generation): discharge piping and expansion joint, antisurge bypass, intake piping and suction drum, air intake, discharge to air, timing gears, speed changers [7,48,51,55–57]. | Air compressor / Steam turbine generator / Turbine admission valve / Turbine drive / Turbine generator brush gear / Compressor platforms | 95–100 / 90–95 / 100 / 95–100 / 95–100 / 90 | Handbook and regulatory database | Miller, 1984 [7] / US EPA, 1971 [18] |
| Steam leaks | Indication of process leaks on corroded pipelines, joints, process valves: High velocity air and steam process leaks [7,48,58]. | Steam leaks (within 25 feet radius) | 100 | Handbook | Miller, 1984 [7] |
| Vents | Intentional and controlled gas or liquid release into atmosphere during emergencies, shut down activities and absence of storage facilities: High velocity air and steam vents [7,48,59]. | Vents (within 10 feet of vent outlet) | 140–160 | Handbook | Miller, 1984 [7] |
| Motors | Power source for driving fans, pumps, generators by converting electric power to mechanical power: Cooling air fan, mechanical and electrical motor noise [7,48,51]. | Motors | 90 | Handbook | Miller, 1984 [7] |

The process equipment listed in Table 3 is inter-connected and inter-dependent resulting in a combination of noise generation mechanisms such as high velocity flow of gases and liquids, cavitation, turbulence, shock and pulsation. The noise emitted by the identified prominent noise sources ranges between 71 dBA up to 160 dBA with vents representing the highest noise emission sources [7]. The proximity of bigger and smaller noise sources adjacent to each other can however mask the noise emitted by the smaller equipment [7,9,22,47,48,50,51]. The reported noise levels in both Tables 1 and 2 were derived and recorded through workplace noise surveys [60,61]. The logarithmic averages of noise levels in Table 3 adds up to reflect those indicated in Table 1 in a case of chemical manufacturing plants.

Noise measurement data for both area and composite measurements resulting from industry noise surveys is voluminous yet remains unpublished. Resources such as the Noise Levels Database developed by the Canadian Centre for Occupational Health and Safety (CCOHS) remain difficult to access for cost reasons [62]. In SA general industry there remains no noise level database, however the Mining Industry Occupational Safety and Health has undertaken the development of the mining industry noise level database [63].

*3.2. Exposure Groups in Chemical Manufacturing Plants*

Chemical manufacturing plants employ different occupational classes which have different noise exposure profiles according to the industry type. Plant equipment used in chemical manufacturing plants require manual operation by these employees, whom by close proximity to these equipments are exposed to the emitted high noise levels [19].

3.2.1. Occupational Classes in Chemical Manufacturing Plants

Various job classification systems exist in the world. According to the ILO occupation classification system, occupations within chemical manufacturing plants are grouped into craft and related trades workers, plant and machine operators and assessmblers [64]. Examples of minor groups of these occupations include electrical mechanics and fitters, electrical line installers and repairers, chemical products plant and machine operators, helpers, plastic machine operators, locomotive engine drivers and related workers, mobile plant operators, steam engine and boiler operators.

In SA, these job categories are clustered in job code 0805 "Chemical, gas, food and beverages production and processing related occupations" [65]. According to OSHA, occupations denoted with standard industrial classification codes 28 to 30 have been historically exposed to excessive noise levels [5].

3.2.2. Relationship between Emitted Noise Levels and Exposure Groups

The exposure groups in chemical manufacturing plants are exposed to both the average and composite noise levels highlighted in Tables 1 and 3. Some job categories, due to proximity to noise sources, task duration and movement patterns in relation to exposure sources, will only be exposed to the average or a fraction of the average noise levels. Examples of the correlation between area noise measurements and employee daily noise dosage are illustrated in Table 4. The daily exposure noise levels are derived and recorded through noise dosimetry surveys where the worker wears a personal noise meter for a task or a job or the full shift [60,66].

**Table 4.** Correlation between activity-based area noise levels and employee daily noise dose.

| Country | Work Activity Observed | Job Category | Area Noise Level Range (in dBA) | Daily Noise Exposure Range (in dBA) | Evaluation Criteria | Reference |
|---|---|---|---|---|---|---|
| UK | Compressed gas supply depot | Workers | 85–94 | 80–90 | Exchange rate: 3 dB, criterion level: 85 dBA, 87 dBA | Institute of Occupational Medicine, 2002 [25] |
| | Paper coating (laminating) | | 81–88 | 84–88 | | |
| | Ship building (blacksmith shop) | | 90–10 | 90–95 | | |
| | Light engineering (fabrication) | | 84–105 | 85–93 | | |
| | Food processing | | 87–94 | 89–92 | | |
| | Coal fired power station | | 93–102 | 85–102 | | |
| | Bottling | | 84–92 | 84–97 | | |
| | Textiles (twisting and winding) | | 88–92 | 85–94 | | |
| | Ferrous foundry | | 81–112 | 86–108 | | |
| | Ship building (heavy fabrication) | | 83–106 | 88–99 | | |
| US | Stator manufacturing | Process operators | - | 84–88 | Exchange rate: 3 dB, criterion level: 85 dBA | NIOSH, 1991 [67] |
| | | Machine tapping operators | | 78–82 | | |
| US | Television manufacturing company: Metal stabilizing lehr | Operators | 81–88 | 83–88 | Exchange rate: 3 dB, criterion level: 85 dBA | NIOSH, 1991 [68] |
| | Frit dispensing department | | 86 | | | |
| | Thump and flush department | | 88 | | | |
| US | Industrial centrifugal manufacturing | Balance machine | - | 85 ** / 88 | NIOSH (Exchange rate: 3 dB, criterion level: 85 dBA) | NIOSH, 1995 [69] |
| | | Basket floor operator | - | 89 ** / 93 | | |
| | | Boring mill (main bay) operator | - | 83 ** / 85 | | |
| | | Crating operator | - | 81 ** / 87 | OSHA (Exchange rate: 5 dB, criterion level: 90 dBA) | |
| | | Fitting floor operator | - | 84 ** / 87 | | |
| | | Welding operator | - | 87 ** / 91 | | |
| Denmark | Various Danish manufacturing industries: Manufacturer of machinery | Workers | - | 81–84 | Exchange rate: 3 dB, criterion level: 85 dBA, 87 dBA | Kock, Andersen, Kolstad, Kofoed-Nielsen, Wiesler and Bonde, 2004 [70] |
| | Manufacturer of furniture | | - | 82–84 | | |
| | Manufacturer of basic metals | | - | 84–87 | | |
| | Manufacturer of wood | | - | 84–86 | | |
| | Manufacturer of minerals | | - | 84–86 | | |
| | Manufacturer of food | | - | 84–86 | | |
| | Manufacturer of fabrication metals | | - | 84–86 | | |
| | Manufacturer of motor vehicles | | - | 83–86 | | |
| | Publishing and printing | | - | 82–84 | | |
| Iran | Textile industry | Spinning operator | - | 93 | Exchange rate: 3 dB, criterion level: 85 dBA, 87 dBA | Nodoushan, Esmaielpour, Ravandi, Mehrparvar and Gholamezadeh, 2008 [71] |
| | | Baling operator | - | 98 | | |
| | | Carding operator | - | 91 | | |
| | | Combing operator | - | 86 | | |
| US | Steel coil manufacturing plant: Pickling and crane cab | Exit labourer 1 | 76–93 | 84 ** / 91 | NIOSH (Exchange rate: 3 dB, criterion level: 85 dBA) | NIOSH, 2017 [72] |
| | | Exit labourer 2 | 50–86 | 70 ** / 84 | OSHA (Exchange rate: 5 dB, criterion level: 90dBA) | |
| India | Glass manufacturing | Coater-1 | - | 92 (83) * | Exchange rate: 3 dB, criterion level: 85 dBA, 87 dBA | Prabu, Gokulram, Magibalam, Senthilkumar and Boopathi, 2018 [73] |
| | | Offline-2 | - | 91 (85) * | | |
| | | Cold end-2 | - | 89 (84) * | | |
| | | Offline-4 | - | 96 (84) * | | |
| US | Urea formaldehyde and Polyurethane foam insulation | Part-time drum washer | 83–94 | 84–92 ^ | OSHA (Exchange rate: 5 dB, criterion level: 90 dBA) | NIOSH, 1983 [34] |
| | | Part-time drum washer | | 88–97 # | | |
| | | Electric-powered screwdriver operator | | 93–96 ^ | | |
| | | Foaming agent drum filling operator | | 95–99 # | | |
| Europe | European petroleum refineries | Crude distillation operators | - | 85–95 ^ / 80–95 # | Exchange rate: 3 dB, criterion level: 85 dBA and 90 dBA | Concawe, 1990 [74] |
| | | Vacuum distillation operators | - | 80–95 ^ / 85–95 # | | |
| | | Isomerisation operators | - | 85–95 ^ / - | | |
| | | Catalytic cracker operators | - | 85–95 ^ / 85–95 # | | |
| | | Catalytic reformer operators | - | 80–95 ^ / 85–95 # | | |
| | | Hydrotreater operators | - | 80–95 ^ / 95 # | | |
| | | Sulphur plant operators | - | 80–95 ^ / 80–95 # | | |
| | | Alkylation plant operators | - | 90–95 ^ / - | | |
| | | Utilities operators | - | 80–95 ^ / 80–95 # | | |

* Values represent before and after engineering control implementation, with after measurements in parenthesis. ^ represent 1982–1984 data. # represent 1985–1988 data. ** Noise levels derived using the OSHA 5 dB exchange rate and criterion level of 90 dBA.

Table 4 data highlights variances in workplace noise levels and employee daily noise dosage which ranges between a minimum of 1 dBA up to a maximum of 10 dBA, where available. Similar to area noise levels, daily noise dosages noted in Table 4 which are at and/or above 85 dBA or 90 dBA represent rating levels at which the employer should implement specific HCP measures to achieve regulatory compliance [10–12].

### 3.2.3. Compensable Claims as Indicator of Noise Exposure

The evidence of noise exposure in the manufacturing sector inclusive of the chemical manufacturing sector is highlighted by the extent of NIHL compensable claims. Figures 1–3 show occupational illnesses including noise induced hearing loss (NIHL) reported to the United States Department of Labour, HSE in the United Kingdom (UK) and the Compensation Commissioner in South Africa [75–78].

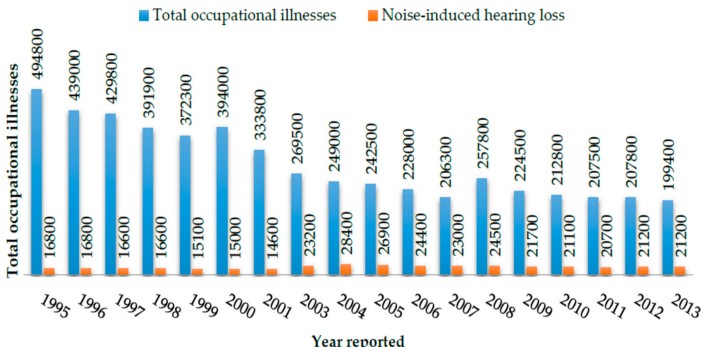

**Figure 1.** Occupational illnesses reported in the United States.

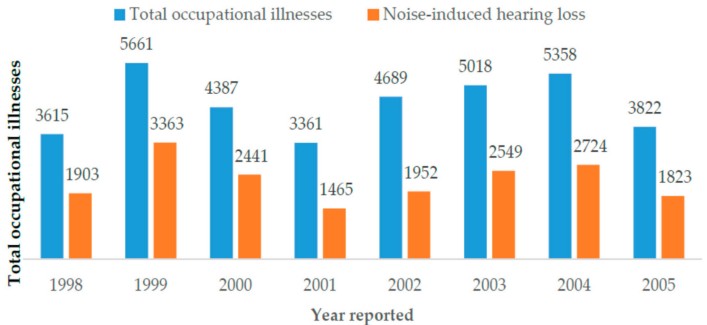

**Figure 2.** Occupational illnesses reported in South Africa.

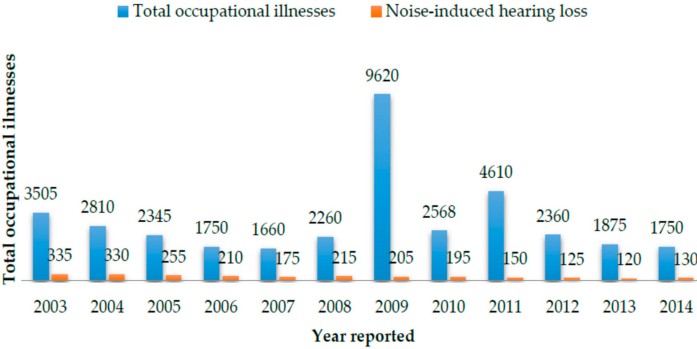

**Figure 3.** Occupational illnesses reported in the United Kingdom.

Figures 1–3 show that NIHL contributed a fair percentage of occupational illnesses between the three countries notwithstanding differences in defining compensable NIHL by each country. In SA, NIHL is by far the highest reported occupational illness. Current NIHL statistics however do not differentiate sectoral origin of each case. In SA, the reported total occupational illnesses exclude those originating from the mining sector whereas those reported in the US and UK statistics include illnesses from the mining sector.

Employees diagnosed with NIHL have a reduced quality of life and limited employment opportunities. These untold consequences place a burden on affected employees. In spite of the

reported NIHL statistics, there is limited knowledge on noise exposure from the workplace across nations [79].

In the US, skin diseases or disorders, respiratory conditions and poisonings amongst others contributed to the total reported illnesses shown in Figure 1.

In SA, other diseases in Figure 2 includes tuberculosis of the lungs in health care workers, pneumoconiosis, dermatitis, occupational asthma, mesothelioma, irritant induced asthma, occupational cancers, chronic obstructive airways diseases, diseases caused by chemical agents and diseases caused by biological agents (excluding tuberculosis).

In the UK, vibration white finger, carpal tunnel syndrome, osteoarthritis of the knee, other musculoskeletal, allergic rhinitis, dermatitis and other musculoskeletal illnesses reported are reflected in Figure 3.

In view of these reported occupational diseases, occupational health and safety regulation to prevent these diseases is therefore justified.

### 3.3. Noise Control in Chemical Manufacturing Plants

Noise regulation mandates employers to implement feasible noise controls through the use of a mixture of engineering controls and administrative controls with HPD use considered as a last resort [10–12]. The noise sources affecting the workforce the most such as those identified in chemical manufacturing plants should receive the highest priority of consideration for engineering control [5,48]. Table 5 shows engineering noise control solutions with projected noise reduction values for prominent noise sources in chemical manufacturing plants. Case histories of the effectiveness of the controls are also shown.

**Table 5.** Noise sources, control solution, approximate noise reduction and case history of effectiveness.

| Noise Source | Noise Control Solution and Typical Noise Reduction | Case History | Noise Control Solution | Noise Reduction Achieved | Author |
|---|---|---|---|---|---|
| Duct and pipe flow | Lagging or acoustical insulation: One-inch thick pipe insulation or double thickness of pipe wall insulation: 20 dB to 40 dB with no acoustical leaks [51,80]. | Steam line regulators generating 97 dBA | Modification of the main valve plug with throttling vanes with a reduction in pressure from 555 to 100 pascal in a 2 inch steam line | Pipeline noise reduced to 85 dBA | NIOSH, 1978 [48] |
| Flow machines (pumps), compressors and turbines | Acoustic enclosure: 5 dB to 10 dB noise reduction for sound insulating wrapping [6]. | Industrial mixer hydraulic pump emitting 97 dBA | Low-cost engineering modifications | Noise reduced to 80 dBA | Advanced Noise Solutions, 2003–2019 [81] |
| | | Screw compressor emitting 100 dBA with tonal content | Reactive silencers fitted into either side of pipe intake | 20 dB overall noise reduction | HSE, 2017 [82] |
| | Acoustic enclosure: 10 dB to 25 dB noise reduction for single shell enclosures with sound absorbing lining. | Steam generator feed pumps emitting high tonal noise at 100–105 dBA, causing noise at the turbine hall to be 92–98 dBA | 4 inch acoustical glass fibre insulating enclosure lined with perforated sheet steel on the inside installed on each feed pump | Turbine hall noise reduction to 88–89 dBA | NIOSH, 1978 [48] |
| | Noise reduction of more than 25 dB for double shell enclosures with sound absorbing lining [83,84]. | | | | |
| Free jets and vents | Silencer: 10 dB to 20 dB noise reduction [6,83,84]. | Ordinary jet noise exceeding 95 dBA | Proper routing of the airstream and installing a silencer | 20 dB noise reduction | Mutchler, 1989 [22] |
| Compressors | Design stage (rotor blade alteration or adjustment control through design): increase in number of rotor blades from 20 to 80 results in 10 dB noise reduction [18,85]. | Motor generator emitting 94 dBA | $\frac{1}{2}$ inch thick glass fibre lined with plywood acoustic enclosure | 10 dB noise reduction | NIOSH, 1978 [48] |
| Gas turbines | Intake and exhaust silencer: a noise insertion loss of 20 to 49 dB in low frequency range and a 40 to 60 dB noise insertion loss at 40 to 60 dB [18,83,84]. | Reciprocating air compressors generating 88 dBA | Intake silencers fitted at each compressor | 17 dB overall noise reduction | HSE, 2017 [82] |
| Machinery noise | Acoustic barriers with absorbent linings: 5 dB–10 dB noise reduction in low frequencies | Gearbox of a 9000 steam turbine emitting 120 dBA inside engine room | Acoustic enclosure using acoustic panels with high transmission properties and a silencer installed at propeller shaft | Noise reduction unknown, noise confined within the acoustic enclosure with a decrease in adjacent areas | NIOSH, 1978 [48] |
| | 20 dB noise reduction in high frequencies [85,86]. | | | | |
| Fans (blowers) | Fan blade design, enclosure and silencers: locating the fan cut-off at optimum clearance in relation to tips of the impeller results in 12 dB noise reduction [18,85]. | High pressure, low-volume centrifugal fan unit emitting 95 dBA | 50 mm thick acoustical panel enclosure | 20 dB noise reduction | HSE, 2017 [82] |
| | | Rotary blowers emitting high noise levels due to unit rotational speed | Hybrid active silencer with absorptive packing for both low and high frequency noise attenuation | 42 dB noise reduction | HSE, 2017 [82] |

The engineering noise control options in Table 5 are intended for application to identified individual noise sources to achieve noise reduction values ranging from 5–60 dB which when correctly applied, would be successful to reduce noise to below the regulated noise rating limit as highlighted by the case histories. Occupational hygienists play a crucial role of advising the process team of the adequacy of these controls [8,46].

Silencer types such as the reactive, reflective, resonator, blow-off and active-adaptive passive provide noise control to airborne noise along the transmission path. Silencers achieve noise reduction by preventing gas pulsation and oscillations at the source, whilst also reducing the conversion of pulsations and oscillations into sound energy [84].

Acoustic insulation when applied to ducts or pipelines involve covering of the outer layer of a pipe with sound-absorbing material to contain the noise within the insulation layer. Acoustic insulation materials are denoted as Class A, B and C depending on the minimum insertion loss requirements which in turn is dependent on the pipe diameter on which they are applied [80].

Engineering noise controls such as acoustic enclosures, silencers and acoustic insulation are cost effective when incorporated during the design phase of new plant installations, purchasing of new machinery and retrospective fitting on existing installations [82,86,87].

The knowledge base on noise engineering control options for noisy equipment operated in chemical manufacturing industries is vast. However, there is limited sharing of this knowledge amongst companies and through research publications [62]. In SA, the literature search could not locate a single article detailing implementation of a noise engineering control measure and its reported effectiveness. In 2019, the *Noise Control Engineering Journal* intends to publish a special issue on case studies of industry implemented noise engineering controls which will in the future become a useful resource for HCP personnel [88].

## 4. Discussion

### 4.1. Noise in the Manufacturing Sector

The broad manufacturing sector utilises an array of plant equipment which emit noise levels as high as 160 dBA. Electric powered motors, an important source of noise exposure, are frequently used at chemical manufacturing plants and during 1974 around two million motors were sold in the US alone [7]. Most manufacturing industries operate plant equipment that emits noise levels exceeding the noise rating limit for hearing conservation. Chemical manufacturing plants also generate high noise levels which vary based on equipment model, size, type, speed of operation and material handled [22]. As an example, compressors used in ammonia plants generate noise levels up to 90 dBA, whereas those used in a low-density polyethylene chemical plant generate noise levels at 100 dBA [56].

### 4.2. Employment and Noise Exposure in the Manufacturing Sector

The manufacturing industry remains the main employer of blue-collar workers, craft workers, plant operators, and machine operators amongst others [1,20,89–92]. The employment statistics in the manufacturing sectors in SA, US and UK are reported at 1.7 million, 16 million and 2.9 million workers respectively [89–91]. According to the 2016 fourth quarter statistics on employment by occupation published by Statistics South Africa, there were 1,977,000 craft workers and related trade, and 1,319,000 plant and machine operators in employment SA for all sectors [91].

In the UK, about 2 million workers are exposed to noise above the lower exposure values (personal noise exposure of 80 dBA) whilst more than a million employees are exposed to noise levels above the exposure action levels [93]. By 2016, the manufacturing sector in the US employed about 15.4 million workers with about 22 million workers are exposed to excessive noise in all industries. Of the 22 million noise-exposed workers in the US, 6 million of these workers were from the manufacturing sector [94–96]. No estimates of the number of employees exposed to noise are currently available in SA's manufacturing sector.

Area noise measurements and noise dosimetry results from the manufacturing sector have been shown to correlate with each other confirming that noise exposes employees to NIHL impacts [97–99]. Differences in noise levels as high as 5 dB are however possible within certain operations between area noise measurements and personal dosimetry as highlighted in Table 4 [99,100].

Available industry NIHL compensation statistics attribute the manufacturing sector as the highest contributor of reported occupational illnesses, totalling about 72% in the US alone in 2010 and 82% in 2007 [89,101]. The NIHL is particularly prevalent amongst specific job categories such as machine operators, plant operators, mechanical fitters amongst others [102]. In South African general industry, NIHL remains the highest compensated occupational illness.

### 4.3. Noise Control for Chemical Manufacturing Plants

Engineering noise control is the ultimate preventative measure for the prevention of NIHL and should be the primary goal of industry implemented HCPs [51,99,103–105]. A systemic review conducted by NIOSH in the US found no field studies evaluating the effectiveness of implemented engineering noise controls. There is thus a great need for publishing noise control measures which have effectively reduced worker noise doses [106]. In a systemic review assessing interventions to prevent occupational NIHL in industry, it was concluded that noise reduction at the source is not sufficiently implemented [107].

Personnel assigned roles of HCP implementation should influence industry managers to protect employee hearing by doing "the right thing" which is to consider and implement noise controls [108]. A non-exhaustive reference of standards that HCP personnel should consider to guide the process of implementing engineering noise controls during any stage of a plant installation are shown in Table 6 along with the scope, stage of consideration and assigned responsibilities.

**Table 6.** Non-exhaustive list of noise control standards and their scope, stage of consideration and responsibility.

| Standard | Scope | Stage of Usefulness/Consideration | | | Responsibility | |
|---|---|---|---|---|---|---|
| | | Design | Replacement | Retrofitting | Supplier | Employer |
| South African National Standard 11688-1. Part 1 [8] | Defines and is intended to provide basic noise control concepts for machinery and equipment during the entire production chain with the aim to assist designers of the final machinery and equipment. | ✓ | ✓ | ✓ | ✓ | ✓ |
| South African National Standard 11690-2. Part 2 [6] | Defines workplace technical aspects relating to noise control. | - | - | - | - | ✓ |
| South African National Standard 11688-2. Part 2 [9] | Defines and is intended for use by machinery and equipment designers and users. Also intended for use by regulators, supervisors or inspectors with the objective of noise reduction in existing plants. | ✓ | ✓ | ✓ | ✓ | ✓ |
| International Organisation for Standardisation. ISO 14163:1998(E) [84] | Defines practical silencer selection for use in gaseous media. | - | ✓ | ✓ | ✓ | ✓ |
| International Organisation for Standardisation. ISO 15667:2000(E) [85] | Defines the acoustical performance and performance criteria of enclosures and cabins. Also defines the agreements of the acoustical and operational requirements between the supplier, manufacturer and end user. | ✓ | ✓ | ✓ | ✓ | ✓ |
| International Organisation for Standardisation. ISO 15664: 2001(E) [109] | Defines noise control design procedures for new plants, plant modifications or during extensions. | ✓ | - | ✓ | ✓ | ✓ |
| International Organisation for Standardisation. ISO 15665:2003(E) [80] | Defines a standardised methodology for the measurement of acoustic performance of Class A, B and C pipe insulation. | ✓ | - | ✓ | ✓ | ✓ |
| International Organisation for Standardisation. ISO 4871 [110] | Provides information on machinery and equipment required declaration of noise emission values and provides a methodology for declared noise emission value verification. | ✓ | ✓ | ✓ | ✓ | ✓ |
| International Electrotechnical Commission. ISO 8297 [111] | Defines an engineering method for determining sound power levels in large industrial plants containing multiple noise sources | - | - | - | - | ✓ |
| International Electrotechnical Commission 60534-8-3. Part 8-3 [112] | Addresses valve and connected piping's noise generated through aerodynamic processes | ✓ | ✓ | ✓ | ✓ | ✓ |
| International Electro Commission 60534-8-4. Part 8-4 [113] | Predicts control valve noise generated noise levels during liquid flow and the resultant noise levels downstream of the valve; and outside the piping. | ✓ | ✓ | - | - | ✓ |
| EEMUA 140 [114] | Indicates the method for specifying maximum acceptable noise levels and describes acceptable test methods for determining equipment noise emission | ✓ | ✓ | ✓ | ✓ | ✓ |
| International Organisation for Standardisation. ISO 11546-2 [115] | Specifies in situ methods to the determination of the sound insulation performance of machine enclosures. | ✓ | ✓ | ✓ | ✓ | ✓ |

Consideration of the guidance or requirements provided in Table 6 ensures that noise is controlled before it is "born" and that due consideration is given during any point of introducing new equipment and machinery into chemical manufacturing plants. In SA, the NIHL Regulations should incorporate the duty of care to force employers' hand to consider equipment listed noise emission levels before introduction into chemical manufacturing plants as it the current case with the control of noise regulations in the UK [10].

### 4.3.1. Workplace Regulation

The health impacts resulting from workplace exposures have resulted in health and safety regulation. Workplace health and safety regulations when fully implemented, result in the reduction of illnesses whilst also giving income security to employees [116]. Although workplace health and safety regulations have not fully eliminated workplace hazards, the implementation of these regulations has reduced workplace injuries and occupational illnesses. The noted reduction in workplace injuries and occupational illnesses is evidence of positive enforcement and inspection outcomes conducted by the regulator through inspectors or compliance safety and health officers [117–119]. Noise regulation compliance means assigning a limit to the hazard level and the imposition of punitive measures in case of non-compliance [118]. The NIHL regulations in SA and the control of noise at work regulations in the UK are examples of noise-specific workplace health and safety regulations. The impact of noise regulation on reducing NIHL cases can however only be measured in the presence of data comparing changes in illness rates for inspected and non-inspected workplaces [120].

### Regulatory Control of Plant Equipment

The legal requirement for the specification of noise emission data of tools, equipment and machinery used in industry is an important regulatory tool in noise reduction efforts for both new and existing plants [121]. Buying quieter plant equipment shifts the focus to equipment designers and manufacturer to meet regulatory noise emission specifications. The buy quiet programme is a NIOSH noise prevention initiative which is composed of (a) a list of existing plant machinery and equipment along with the listed noise levels; (b) company policy statement committing to buying quiet; (c) materials and tools for educating and promoting the buy quiet programme; and (d) cost–benefit analysis for buying quiet. In the buy quiet programme, manufacturers are encouraged to design quieter equipment and machinery whilst companies are also encouraged to buy or rent equipment which is quieter [122]. Noise emission data specification related legislation has however been largely ignored and both equipment designers and purchasers are said to only pay lip service to the requirement [18]. The noise ranges extrapolated on Table 2 attest to this fact.

Additional requirements that can be built into the noise data sheets requirements for machinery include placing additional restrictions on machinery containing tonal and impulsive noise components. Plant owners have to consider the cost effectiveness of selecting machinery with low noise emission levels during the design phase of a plant or selection of new machinery [109]. Due to the interconnectedness of machinery in chemical manufacturing plants, noise prediction incorporating all noise emitting machinery should be done during the design phase of new installations [80,84,85,109].

### Regulatory Control through Inspection and Enforcement of Noise Regulations

Noise regulations are intended for worker protection in the workplace and are implemented by the employer whilst the regulator conducts inspection and enforcement [123]. The inspection and enforcement are conducted to ensure legal compliance to health standards [124]. The inspectors use preventative tools such as risk evaluation, promotion of best practices, information and awareness campaigns, guidance and sanctions during inspections to secure compliance to health and safety regulations [125].

Noise regulations are intended for worker protection in the workplace and are implemented by the employer whilst the regulator conducts inspection and enforcement [123]. The inspection and

enforcement are conducted to ensure legal compliance to health standards [124]. The inspectors use preventative tools such as risk evaluation, promotion of best practices, information and awareness campaigns, guidance and sanctions during inspections to secure compliance to health and safety regulation [125].

Between October 2015 and September 2016, OSHA issued 691 citations for 370 inspections conducted with penalties amounting to $1,665,895 for the violation of the noise standard. The chemical manufacturing sector received 20 citations in 10 inspections with penalties amounting to $105,897 whilst the plastics and rubber producers were issued with 35 citations for 18 inspections with penalties amounting to $157,303. [126]. Although evidence of enforcement and inspection in SA exists, the DEL annual report does not detail the specific violations noted during enforcement and inspections. In the 2015/2016 reporting year, 20,476 notices (citations) were issued to industry for health and safety violations during 23,678 workplace inspections [127]. No inspection and enforcement data relating to the citations related to the noise standard is available for the UK.

To highlight the important role of inspection and enforcement, inspections conducted by OSHA inspectors in 1973 showed that occupational illnesses and injuries were reduced by an estimated 16 percent. Inspections conducted in 1974 however showed no statistically significant inspection impact [128]. The results of studies attempting to measure the effectiveness of OSHA enforcement however, remain mixed but point to the positive impact on the reduction of incidence of occupational illnesses and injuries [129]. Incomplete enforcement of health and safety standards is identified as a leading factor for low impact inspection outcomes [120].

Interestingly, inspections coupled with penalties conducted by OSHA are reported to have resulted in a reduction of 19 percent lost workday injuries by 1979–1985. This percentage reduced to 11 in 1987–1991 [130]. An increase in the number of health and safety inspections conducted in the US' manufacturing sector has also been shown to result in a decrease in the number of citations and worker exposure levels, further highlighting this need [131]. However, increased enforcement and inspection in the current economic climate adds additional fiscal pressure on government budgets [132].

Organisational and Administrative Controls

Similar to the mining industry, there remains limited or no evidence of the implementation of administrative controls to reduce worker exposure in the manufacturing industry [133]. The organisational aspects of noise control include consideration of work methods which generate low noise levels. The organisation of work to reduce noise considers limiting exposure duration by reducing exposed employees to an absolute minimum, scheduling work activities exposing few employees accompanied by rest periods away from noise sources [134]. The reliance on administrative controls should however be accompanied by continuous checks to ensure compliance and correct application [73]. Administrative controls should incorporate the use of worker dosimetry, time-motion studies and equipment noise profiling to increase effective utilisation. Despite lack of their economic feasibility, an apparent advantage of administrative controls is that they have a low cost base and consume less time during implementation compared to engineering controls [133].

Hearing Protection Devices

Noise regulations also prescribe the transient use of HPDs for exposure control. However, HPDs can only be effective where users are properly trained in their use and correctly selected based on the country's preferred rating method to avoid misinterpretations in selection outcomes [33]. Compared to other HCP elements, HPDs have been found to be the most implemented element [106].

Notwithstanding any degree of protection afforded by wearing HPDs, their real-world performance has been shown to be highly compromised in many studies, which further highlights the need for implementing engineering noise control [135].

Noise in chemical manufacturing plants has not been eliminated despite regulatory enforcement, inspection and HCP institution [136]. The HCPs implemented by industry should include additional

aspects such as allowing for noise prediction techniques for new plant installations which are available to assist employers in noise reduction efforts [8,48,87,137].

### 4.3.2. Costs and Benefits of Noise Control and Regulation

The implementation of noise control and other aspects of noise regulations place a financial and compliance burdens on employers. However, this should not discourage employers from implementing feasible engineering noise controls. The annual costs for the manufacturing sector in the US relating to compliance with the occupational noise standard in 1993 were estimated by OSHA to be around $210.3 million. The compliance costs for each enterprise in the sector were also estimated at $638 thousand based on the 1993 dollar terms. On the other hand, the costs related to HPDs required for compliance with the same standard were estimated at $34.2 million. The overall estimated compliance costs to OSHA-related regulations in the US in 1993 were estimated between $23.1 billion and $46.7 billion [138].

In the UK, the final regulatory impact assessment of the control of noise at work regulations 2005, also estimated compliance costs segregated into costs related to familiarisation, assessments, information, preparation of programme, reducing exposure, HPDs, signage and audiometric testing. On the first year on implementation, employer costs were estimated between £117 million to £202.6 million, whilst the 10 year costs were estimated to be between £477.6 million to £676.3 million. The highest cost item predictably related to the implementation of engineering controls was estimated at between £27.5 million to £109.9 million in the first year, and increased to between £65.2 million to £268 million for the 10-year cost period. The HSE estimates employer costs for each noise-exposed worker at £35.60. [139].

There however remains limited studies detailing the exact overall costs of compliance to all occupational health and safety regulations [132]. This is true for SA where there are no publicly available cost estimates and benefits of implementing occupational health and safety regulations inclusive of the NIHL regulations. Notwithstanding the costs relating to compliance to the NIHL regulations, employers and the government in SA should play their legal roles of ensuring that the good intentions of these regulations are met. Afterall, these regulations were enacted into law following extensive deliberations and agreements amongst the tripartite structure inclusive of government, labour and unions.

The benefits of noise control and regulation are however not immediately visible due to the delayed onset of NIHL. However, an ancillary and tangible benefit of noise control and regulation is that it highlights the important role played by professions such as occupational hygienists in workplace health and safety programmes [140]. The benefit of a noise control measure to a worker however requires noise quantification through dosimetry measurements before and after the intervention [141]. In general, the costs of occupational safety and health actions are easily visible compared to benefits which tend to be underestimated. Additionally, the return on investment related to occupational health and safety compliance initiatives is influenced by an enterprise's current practice, implying that an enterprise with a dismal practice in noise control and prevention will notice major spinoffs for a small investment made to address the same issues [142].

## 5. Conclusions

Chemical manufacturing plants, and the manufacturing sector in general, operate plant equipment and machinery that emit noise levels exceeding regulated exposure levels. The US manufacturing sector is the largest when compared to the UK or SA, and has the highest number of noise-exposed employees. Within the chemical and petroleum industry, refinery units emit the highest average noise levels. In terms of specific noise sources within chemical manufacturing plants, vents emit the highest noise levels.

The manufacturing sector inclusive of the chemical manufacturing industry is a major source of employment for different occupations such as process operators and maintenance personnel, who are

inherently exposed to the emitted noise. The majority of NIHL cases reported in SA emanates from the manufacturing sector whereas the sector also contributes a sizeable number of these cases in the US. In the UK, NIHL cases are low as a proportion of total occupational illnesses reported.

For the noise sources identified in chemical manufacturing plants, there exists alternative engineering noise control solutions for each source. Case studies of implemented engineering noise control measures show that engineering control remains the only option for reducing noise levels, at the point of generation, to below the regulatory levels.

Regulatory initiatives such as noise regulations and regulation of machinery have seemingly not had a tangible effect on noise control at the workplace. Increased inspections and enforcement by the regulatory authorities can help arrest the continuation of the status quo. The South African inspection and enforcement regime relating to noise remains vague compared to that used by OSHA in the US. In general, the UK's noise control efforts through regulation have seemingly been more effective than those employed in SA and the US based on the NIHL statistics highlighted in this paper. The cost of achieving compliance with the noise regulations will vary based on the number of noise-exposed employees per enterprise [140]. The greatest expenditure in health and safety programmes relates to training and personal protective equipment provision but excludes health and safety personnel costs [143,144]. The investments made in the preventative aspects of health and safety such as training, enforcement and inspection, and medical surveillance are however not easily isolated when determining the cost–benefit of regulations [145]. For enterprises, decision makers are therefore required to consider the enterprises' fiscal position when selecting and prioritising noise control measures for implementation [146].

In SA, a new approach can be to consider the adoption of new noise regulatory levels for both existing and new plant installations. As an example, new installations should be regulated on a 75 dBA rating level, whereas existing plants can be regulated based on setting continuous targeting of noise reduction up to 80 dBA. On a regulatory level, SA can also do well by clearly defining conditions required for demonstrating legal compliance with the NIHL regulations by inclusion in the DEL inspection manual, as is the case in the US. The DEL in SA can also consider adopting the Department of Mineral Resources' approach of setting short-term noise reduction targets for identified noise sources.

Another important aspect that can be considered in the SA regulatory environment is the formalisation of a peak noise-rating limit as is the case in the UK, which is regulated at 140 dBC. The current 85 dBA noise rating limit assumes that workers are only exposed to continuous type of noise.

Although the enforcement and inspection regime in SA is largely self-regulatory, the DEL can also initiate initiatives such as the buy quiet programme and incentivising industry for implementing effective noise control, approaches already advocated by NIOSH in the US.

There exists a body of knowledge relating to noise exposure, noise sources, exposure groups and control options for chemical manufacturing plants. This knowledge should be harnessed by all stakeholders involved in various stages of HCPs, enforcement and inspection to achieve quieter workplaces.

The future success of workplace health and safety relies on governments, researchers, employers, employees and health and safety officers' strong commitment [119]. Thus, in SA, studies addressing the socio-economic impact of occupational diseases including NIHL and the evaluation of the effectiveness of the noise regulations are advocated to ensure that the prevention of this workplace scourge is prioritised [147]. Research is also required in the area of compliance costs related to the NIHL regulations [138]. Other research areas include studies focusing on the implementation procedures and evaluation methods of administrative controls for reducing worker noise exposures [134].

**Author Contributions:** O.R. conceptualised the study, conducted the literature study, drafted and edited the manuscript. J.C.E. and J.L.H. reviewed the technical content and layout of the manuscript. All authors read and approved the final manuscript.

**Funding:** This research received no external funding.

**Acknowledgments:** The primary author acknowledges Karabo Shale, Cape Peninsula University of Technology, Faculty of Applied Sciences, Department of Environmental and Occupational Studies, for his encouragement and support during the initial part of the broader project.

**Conflicts of Interest:** The authors declare no conflict of interest.

**Ethical Statement:** The results presented in this paper form part of a broader study for which ethical clearance was obtained from the Tshwane University of Technology (TUT) Ethics Committee (FCRE 2016/03/012 (SCI)).

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
