# Peer review of "Noise Sources and Control, and Exposure Groups in Chemical Manufacturing Plants"

_applsci, doi:10.3390/app9173523_

Round 1

Reviewer 1 Report

The paper is a review about noise emissions in chemical plants. Noise exposure is a relevant topic deserving attention, and I trust authors when they say that there is no specific study in noise inside chemical plants.  However, the current year is 2019 and, as reported by the authors themselves, the research is performed in 2016. This make the “review” paper unpublishable a priori. I kindly ask the authors to spent time updating their work to 2019. Furthermore, I kindly suggest the authors to specify better that 85 dB(A) is the value of exposure to noise in working environments, not noise emission! Noise emission and noise exposure is a very distinct physical entity.

Reviewer 2 Report

This submission is entitled as "A review" and its article type is Review, however, although the study is based on literature survey, it leads to a conclusion to raise a potential noise problem in a specific country. The survey is comprehensive and the discussion is reasonably given, this reviewer basically agree to publish this manuscript. Therefore, the comments from this reviewer are rather few and only minor points suggesting the authors to reconsider before publication:

The methodology: The work is based on literature survey using internet databases. It would be helpful for readers to give more detailed explanation, e.g., used search words, publication date, etc.

"Ethical statement" may be better to be moved to the end of the article.

Table 1 does not convey the information about the country / region, which may be of some interests.

Page 11, Discussion: There is a statement that noise level of "160dB". It is not realistic. Check if this is true. Also, used frequency weighting should be added, e.g., dB(A), dB(C), etc.

Conclusions raises the country specific problem. This is encouraged to present the problem. However, also some more general conclusions could be detailed. (If the purpose of this article is review the state of the art, it is very important.)

Reviewer 3 Report

The review is well written, it presents major problems of noise exposure and control at the chemical industry. It also shows a good comparison of the situation in UK, USA and SA. The problem is - as many times noted also elsewhere - that technical noise control is overlooked and focus is very often in HCP and HPD. Therefore, this review does not give a new approach. On the other side, there are not many articles written on this item. References are of good quality and the literature search is good

Some minor comments:

Lines 221-222 which table, where are the highlighted values?

Line 247 guiet?

Lines 286-288 necessary? Is it a conclusion from this review?

Round 2

Reviewer 1 Report

Now at line 55 is reported that the literature review is performed between 2016 and 2019, forgetting about what was before my previous suggestion.

Pleas check the paper in details, because some formatting issues are presents according to journal editorial system. Furthermore, control editing such as putting .0 after numbers.

Please insert, in the introduction, some lines more in order to better introduce the importance of noise issue on health. This will help the reader to better understand the importance of noise expusure prevention.
